# Functional Analysis of Antipsychotics in Human iPSC-Based Neural Progenitor 2D and 3D Schizophrenia Models

**DOI:** 10.3390/ijms26094444

**Published:** 2025-05-07

**Authors:** Kiara Gitta Farkas, Katalin Vincze, Csongor Tordai, Ece İlay Özgen, Derin Gürler, Vera Deli, Julianna Lilienberg, Zsuzsa Erdei, Balázs Sarkadi, János Miklós Réthelyi, Ágota Apáti

**Affiliations:** 1Institute of Molecular Life Sciences, HUN-REN Research Centre for Natural Sciences, H-1117 Budapest, Hungary; 2Doctoral School, Semmelweis University, H-1094 Budapest, Hungary; 3Department of Molecular Biology and Genetics, Bilkent University, 06800 Ankara, Turkey; 4Salus Ltd., H-1037 Budapest, Hungary; 5Department of Psychiatry and Psychotherapy, Semmelweis University, H-1083 Budapest, Hungary

**Keywords:** schizophrenia model, induced pluripotent stem cell-derived neural cells, three-dimensional spheroid, antipsychotic medication, cell adhesion

## Abstract

Schizophrenia is a complex psychiatric disorder of complex etiology. Despite decades of antipsychotic drug development and treatment, the mechanisms underlying cellular drug effects remain incompletely understood. Induced pluripotent stem cell (iPSC)-based disease and pharmacological modelling offer new avenues for drug development. In this study, we explored the development of two- and three-dimensional neural progenitor cultures and the impact of different antipsychotics in a schizophrenia model. Four human iPSC lines, including two carrying a de novo *ZMYND11* gene mutation associated with schizophrenia, were differentiated into hippocampal neural progenitor cells (NPCs), cultured either in monolayers or as 3D spheroids. While in monolayers the proliferation of the NPCs was similar, spheroids showed significant differences in scattered cell number and outgrowth size between schizophrenia mutant and wild-type NPCs. Since there is only limited information about the effects of antipsychotic agents on neural progenitor cell proliferation and differentiation, we investigated the effects of three molecules, representing three subgroups of antipsychotics, in the 2D and 3D NPC models. Our findings suggest that cell adhesion may play a crucial role in the molecular disease pathways of schizophrenia, highlighting the value of spheroid models for mechanistic and drug development studies. These studies may significantly help our understanding of the effects of schizophrenia on neural development and the response of progenitors to antipsychotic medications.

## 1. Introduction

Schizophrenia (SCZ) is a complex, chronic psychiatric disorder with a polygenic background, affecting approximately 1% of the adult population globally [1]. The disorder presents with symptoms that pose significant challenges to social functioning, educational attainment, and work performance, despite the availability of pharmacological and psychosocial interventions. Extensive research into the genetics and neurobiology of SCZ has uncovered various molecular mechanisms and brain alterations associated with the disorder [2]. The pathogenesis of SCZ is multifaceted, including theories of dopaminergic dysregulation and glutamatergic abnormalities [3], primarily affecting the striatum and cortex. Recent neuroimaging studies have also identified structural and functional abnormalities in the hippocampus of SCZ patients [4,5], suggesting that increased extracellular glutamate concentration and dysregulated glutamate neurotransmission may contribute to these abnormalities [6]. To advance our understanding of SCZ, a variety of methodological approaches have been employed. While post-mortem histopathological studies have revealed significant abnormalities [7], they are limited in distinguishing the effects of medications from disease-related pathophysiological processes. Technologies such as functional magnetic resonance imaging (fMRI) and positron emission tomography (PET) facilitate studies in living subjects, though they have limitations in spatial and temporal resolution [8]. The advent of human-induced pluripotent stem cells (hiPSCs) has opened new avenues for research [9]. In our previous work, we successfully reprogrammed blood cells from a schizophrenia patient carrying a de novo mutation in the *ZMYND11* gene into hiPSCs (Pat-Mut) [10]. The mutation was corrected in the patient-derived iPSC line using CRISPR editing (Pat-Wt) and introduced into the wild-type control line (Ctrl-Mut). Along with wild-type control line (Ctrl-Wt), these four hiPSCs were differentiated into hippocampal neural progenitor cells (NPCs) and dentate gyrus granule cells (DGGCs) using a hippocampal differentiation protocol [11]. Transcriptomic analyses revealed significant differences in genes related to neuronal differentiation, cell- and matrix-adhesion, and synaptic function, indicating altered developmental pathways. Functional studies revealed reduced glutamate reactivity in DGGCs; however, NPCs were not characterized functionally in this work.

The development of antipsychotic medications, the primary treatment for schizophrenia, has evolved through three generations. The first generation was primarily developed by antagonizing dopaminergic receptors, specifically targeting dopamine D2 receptors to exert antipsychotic effects. The second generation of antipsychotics, i.e., atypical antipsychotics, operate through a dual blockade of both 5-hydroxytryptamine (serotonin) and dopamine receptors. With the third generation, the therapeutic focus for treating schizophrenia has expanded beyond D2 receptor blockade to include D2 and D3 receptor partial agonism and the exploration of additional targets such as 5-HT1A, 5-HT7, and mGlu2/3 receptors. The primary advantages of second- and third-generation antipsychotics over their first-generation counterparts include a reduction in extrapyramidal side effects and an improvement in negative symptoms. Third-generation antipsychotics do not block D2 receptors; rather, they act by partial agonism, thus the modulation of the dopamine neurotransmitter system remains a crucial aspect of their antipsychotic action [12].

Despite the widespread use of these psychotropic medications, there is limited information about their effects on human neural progenitor cell proliferation and neural differentiation. In our previous study, we investigated the effects of three antipsychotic agents (first-generation: haloperidol and second-generation: olanzapine and risperidone) on the function and differentiation characteristics of neural stem cells derived from healthy, human-induced pluripotent stem cells. Our results indicated that short-term treatments primarily affected neurite development rather than cell number (except for haloperidol) or the differentiation ability of neuronal progenitors [13].

In this work, we integrate the findings of our previous studies to investigate the effect of antipsychotics on hippocampal NPCs carrying the ZMYND11 mutation associated with schizophrenia. Our experiments examine the effects of first- (haloperidol; HP), second- (risperidone; RP), and third-generation (aripiprazole; AP) antipsychotics, employing both two-dimensional and three-dimensional studies to better characterize the phenotypes.

## 2. Results

### 2.1. Characterization of 2D NPCs Differentiated from ZMYND11 Wild Type and Mutant iPSCs

In our previous publication [10], we focused on hippocampal neurons derived from iPSCs of a schizophrenic patient carrying a de novo mutation in the ZMYND11 gene. We examined NPCs only at the mRNA level, where we found significant differences between mutant and wild-type NPCs, particularly concerning cell adhesion processes. Therefore, we decided it worthwhile to examine these NPCs in more detail. Cell adhesion affects cell migration and division, so we first examined NPCs in a 2D model with a cell proliferation assay and a wound healing assay. We differentiated four types of iPSCs (Pat-Mut, which originated from a schizophrenic patient, Pat-Wt, where the patient’s de novo mutation in the *ZMYND11* gene was corrected, and an independent healthy control cell (Ctrl-Wt) or its *ZMYND 11* mutant version (Ctrl-Mut) into hippocampal NPCs (see methods for details). The NPCs were seeded into polyornithine-laminin-coated plates, one part was allowed to divide, the other part was allowed to be almost confluent and then scratched. The endpoint of the proliferation assay was day 3 (Figure 1a), as after that the cultures became so dense that we could not reliably evaluate the data with the analysis programme. The scratch assay was evaluated after 24 h, as in several cases the scratch was closed in the 48 h samples (indicating efficient migration and proliferation of NPCs) (Figure 1b and Appendix A). No significant difference was found in cell division between patient-derived cells, while in the case of control-derived cells, Ctrl-Mut NPCs grew more slowly than Ctrl-Wt (Figure 1c). In the case of scratch closure, the control-derived cells showed no difference, but the Pat-Mut cell line closed the wound significantly faster than the Pat-Wt line (Figure 1d).

### 2.2. Receptor Expression and Proliferation of 2D NPCs Differentiated from ZMYND11 Wild Type and Mutant iPSCs After Antipsychotic Treatment

In the next experiment, we aimed to investigate how antipsychotics used in clinical practice affect the growth of NPCs. Analyzing the mRNA sequencing data reported in our previous article [10], we found expression of the D2, D4 dopaminergic and 5HT 2A, 5HT 2C, and 5HT 7 serotonin receptors among the receptors involved in the action of antipsychotics (Figure 2a). We subsequently treated the NPCs with both lower and higher concentrations of a first-generation antipsychotic (haloperidol), a second-generation antipsychotic (risperidone), and a third-generation antipsychotic (aripiprazole). In no case did the antipsychotics change the proliferation of NPCs under these conditions (Figure 2b).

### 2.3. Spheroid Formation of NPCs Differentiated from ZMYND11 Wild Type and Mutant iPSCs

Given that NPCs reside in a three-dimensional environment in vivo, we sought to investigate their behaviour within a 3D context. To this end, we cultured the NPCs into spheroids to better mimic their native conditions, using progenitor-specific medium (NPC medium) to prevent differentiation into neurons. When an equal number of NPCs were plated on the ultra-low attachment (ULA) plate, a significant difference in the size of mutant and wt spheroids was observed already at this stage (Figure 3a). When the spheroids were measured on day 3, regardless of the overall genetic background, the mutant spheroids were significantly larger than their wt counterparts (Figure 3b). This may be at least partly because the spheroids derived from the NPCs carrying the mutation organized into a single spheroid, while smaller, separate clumps often appeared next to the wt spheroids (Figure 3a, white arrows). When quantified, the number of separate clumps was five vs. 91 between Ctrl-Mut and Ctrl-Wt, and 0 vs. 73 between Pat-Mut and Pat-Wt.

### 2.4. Spheroid Outgrowth of NPCs Differentiated from ZMYND11 Wild Type and Mutant iPSCs

After three days in suspension culture, the spheroids were transferred to polyornithine-laminin-coated plates for attachment. Further, when the spheroids were maintained in NPC medium, a clear difference was observed between the mutant and wt spheroids on day 3 after attachment (Figure 4a,b). Both the number of cells spreading from the spheroids and the extent of outgrowth were significantly reduced in the mutant spheroids when compared to their wt counterparts (Figure 4c,d).

### 2.5. Effect of Antipsychotic Treatments on Spheroid Outgrowth of NPCs Differentiated from ZMYND11 Wild Type and Mutant iPSCs

When the adherent spheroids were treated with antipsychotics for 3 days (Figure 5 and Figure 6), we found that in both Ctrl-Wt and Pat-Wt, all treatments reduced the number of cells spreading from the spheroids and the size of the outgrowth. In the case of NPCs differentiated from patients (Pat-Mut), high concentrations of HP significantly reduced the number of cells spreading from the spheroids. Certain treatment conditions, although not reaching statistical significance, resulted in an increased number of cells spreading from the spheroids (e.g., low concentrations of RP, as shown in Figure 5) or an increase in the size of the outgrowth (e.g., low concentrations of AP, as depicted in Figure 6).

### 2.6. Spheroid Outgrowth of NPCs During Neural Differentiation and the Effect of Antipsychotic Treatments

The characteristics of NPCs may play a role in the emerging phenotype not only during the growth and migration stages, but also during the differentiation phases. In the next experiments, we focused on the initial stage of differentiation (4 days—primarily because the adhered spheroids, especially in the mutant cases, detached from the surface when cultured for longer periods). The attached NPC spheroids were maintained in differentiation medium for 4 days, and then processes and cell bodies were visualized using Calcein-AM staining. Again, the morphological difference between mutant and wt was very striking, regardless of the genetic background (Figure 7a). The long processes of the spheroids carrying the mutation did not show any branches, and there were hardly any scattered cells on them, while the wild-type spheroids developed complex networks of processes, on which there were also many scattered cells (see also Appendix A).

Unfortunately, the available analysis programmes were not suitable for quantifying this phenotypic difference, and the features that we finally used (the furthest outgrowth distance and the average outgrowth distance) showed no difference between the untreated (Figure 7b,c) and treated cells (Figure 8).

### 2.7. Characterization of Cell Adhesion Based on mRNA Sequencing Data

Our previous publication [10] presented mRNA sequencing data for the same NPCs, revealing that in mutant NPCs, underexpressed genes were associated with the extracellular matrix, cell adhesion, and cell–extracellular matrix connection terms, according to Gene Ontology (GO) enrichment analysis. Indeed, in spheroids maintained in NPC medium, both the number of spreading cells and the extent of outgrowth were significantly reduced in mutant spheroids, compared to their wild-type counterparts (Figure 4c,d). The observed phenomenon may be attributed to the reduced cell adhesion and migration capacity of the NPCs in mutant spheroids. To further analyze this observation, we plotted the GO terms appearing in Pat-Mut and Ctrl-Mut NPCs in a Venn diagram (Figure 9a). In addition to the common intersection, more GO terms appear in patient-derived NPCs, indicating that the reduced migration capacity observed here is not exclusively caused by the *ZMYND11* mutation. The most frequently occurring genes in GO terms are *TGFB2, COL1A1, MMP14*, and *HAS2*, all of which are related to cell adhesion and cell migration (Figure 9b). Underexpression of these genes may greatly affect cell–cell and cell–matrix interactions.

## 3. Discussion

iPSC modelling of neuropsychiatric disorders is becoming more popular by offering a valuable alternative to postmortem brain studies and animal models, enabling the investigation of selected neural cell functions in human samples [14,15]. The influence of donor genetic background and iPSC clonal variations can serve as confounding factors, suggesting a need for larger sample sizes. Functional differences often emerge only under specific cellular states, such as neural activation or progenitor proliferation. These neural cultures primarily replicate early neurodevelopmental processes, making them particularly suitable for modelling psychiatric disorders with neurodevelopmental components, like schizophrenia [16,17]. Another reason why the study of hippocampal progenitors may provide valuable data for schizophrenia research is that alterations in adult neurogenesis in the subgranular zone of the hippocampus may be a potential cause of SCZ [18]. Research findings suggest that postmortem hippocampal samples from SCZ patients show a reduced expression of Ki-67, a cell proliferation marker [19], and that SCZ patients have smaller hippocampal volumes based on a meta-analysis of structural MRI studies [20]. Furthermore, some studies have shown that clinical improvement is accompanied by normalization of hippocampal size [21]. In addition to changes in cell proliferation, impaired maturation of adult dentate granule cells has also been reported in SCZ patients [18].

Moreover, antipsychotic medications are often prescribed to treat people with a wide range of psychiatric conditions, including schizophrenia, bipolar disorder, depression, anxiety, and personality disorders. Commonly, second-generation antipsychotic medications are selected for pregnant women. Most antipsychotics cross the placenta [22] and can cause developmental disorders [23], so the study of neural progenitors may be of particular importance in elucidating the mechanisms of unwanted side effects and in the selection of personalized medication. A current study concludes that maternal exposure to clinically relevant doses of risperidone may induce neurostructural changes in the developing hippocampus and striatum, and cognitive sequelae in young rat offspring, respectively [24].

In this study, we examined a single gene mutation (*ZMYND11*) across different genetic backgrounds (Pat and Ctrl) in hiPSC-derived hippocampal progenitors at a specific neural development stage. To better characterize the emerging phenotypes, we employed multiple functional assays and utilized both 2D and 3D culture models.

Although 2D culturing has the advantage of being relatively homogeneous and cellular phenotypes can be easily examined, it does not resemble the in vivo neural environment, and the results obtained are often contradictory. On the other hand, the anatomical and morphological similarity of 3D brain organoids to the developing human brain makes them excellent models for studying schizophrenia. Cortical organoids have already been utilized to model SCZ [25,26]. However, while these studies demonstrate compelling transcriptomic and phenotypic changes in SCZ organoids, caution is necessary due to morphological variability and uneven cellular compositions across organoids.

Three-dimensional spheroids are similar to 2D cultures in that they have a more homogeneous cell composition, but similar to organoids, they more closely resemble in vivo conditions. Spheroids have been extensively used for studying tumours [27,28,29] or for the generation of specific organoids of different brain regions from stem cells [30,31], and as a starting point for different brain assembloids [32]. In the above-mentioned works, the 3D technique was used to create more mature neurons and stronger connections. In our present work, we focused on studying neuronal cells that were still in the progenitor state, thus better mimicking cell–cell and cell–matrix interactions.

In our two-dimensional culturing study, results from cell division and scratch assays were inconclusive due to varying trends across different genetic backgrounds. In the proliferation assay, Pat-Mut cells outperformed Pat-Wt, whereas Ctrl-Wt cells exhibited significantly faster growth than Ctrl-Mut. Conversely, in the scratch assay, Pat-Mut cells closed the wound faster than wild-type, a trend not observed in the Ctrl background. Although CRISPR-modified cells generally grew slower, this was not significant for Pat-Wt. Previous data in mouse embryonic hippocampal cultures showed that 72 h treatment with HP and OL inhibited proliferation [33], and a mild cytotoxicity of risperidone is also known from the literature. However, these effects emerged at approximately 100 times higher concentrations than those applied here [34,35].

According to our previous findings, using hippocampal NPCs obtained from fibroblast-derived iPSCs [13], the lack of significant differences in cell numbers following antipsychotic treatment was expected. Receptor expression in these NPCs mirrors was previously reported in vitro human data [13,36]. In the hippocampal NPCs studied here, although D4 expression predominates, D2 expression is also present, and 5HT 2A, 5HT 2C, and 5HT 7 serotonin receptors are also expressed among the receptors affected by antipsychotics. In 6-week-old neurons differentiated from NPCs, the receptor distribution is similar (Appendix A), with some receptor subtypes (such as 5 HT 1E or 5 HT 5A) being expressed only in neurons. In general, the expression level in neurons is higher than in the progenitor state, which may reflect the maturation process. The toxicity observed in previous publications (in the case of HP and RP) may be due to the different cell type, antipsychotic concentration, or measurement technique.

To mimic cell–cell and cell–matrix interactions is particularly important if the expected differences between disease-model and healthy samples are related to cell adhesion and extracellular matrix composition differences. Mutant spheroids showed differences in spheroid size, number of cells migrating from the spheroid, and migration distance compared to wt, regardless of genetic background. However, the fact that the differences observed in Ctrl-Mut and Ctrl-Wt spheroids are smaller than in patient-derived spheroids (Pat-Mut vs. Pat-Wt) also suggests a difference in genetic background, although this does not diminish the significance of the observation that the migration defect appears in mutant NPCs regardless of genetic background. The most frequently occurring genes in GO terms are related to cell adhesion and cell migration (Figure 9b). TGFB2 (Transforming Growth Factor Beta 2) is a cytokine that plays a significant role in cell growth, differentiation, and repair processes [37,38]. *COL1A1* encodes the alpha-1 chain of type I collagen, a major structural protein in the extracellular matrix. TGFB2 can regulate the expression of collagen genes, including COL1A1. TGFB2 is known to stimulate the production of collagen and other extracellular matrix components, thereby influencing tissue remodelling and repair [39]. MMP14, also known as membrane-type matrix metalloproteinase 14 (MT1-MMP), is an enzyme that plays a crucial role in the remodelling of extracellular matrix (ECM). Connection of MMP14 to cell adhesion is primarily through its ability to degrade various components of the ECM, which may influence the adhesion properties of cells [40]. HAS2, or hyaluronan synthase 2 [41], is closely connected to cell adhesion through its role in synthesizing hyaluronan, which influences the ECM structure, interacts with cell surface receptors, and modulates cellular behaviours critical for adhesion and migration. The further study of these signalling pathways may lead to a better understanding of the mechanism and the role of cell adhesion in schizophrenia. The present observation that differentiating spheroids, particularly those harbouring the mutation, do not remain attached to the surface after 10 days, further underscores the possibility that the low expression levels of adhesion molecules in mutant cells may contribute to the development of the disorder.

Various models are available to study the pharmacological effects of different antipsychotics (reviewed by [42]); however, testing these drugs in human iPSC-derived cell lines remains a significant gap in the current literature. There is a scarcity of well-designed pharmacological studies examining the effects of antipsychotics in human iPSC-derived neurons. We have shown that while antipsychotic treatment did not affect NPC growth in 2D cultures, a 3-day treatment in 3D spheroids resulted in a significant decrease in both the scattered cell number and the size of outgrowth in wild-type cells in all cases. In contrast, mutant NPCs exhibited a more complex response, with some treatments increasing and others decreasing these indicators. These data suggest that certain antipsychotics may enhance the outgrowth of cell numbers and sizes in mutants, potentially narrowing the gap compared to wild-type cells.

As a summary, in this study, we compared NPCs with two different genetic backgrounds, carrying de novo mutations identified in a patient. In all cases, NPCs carrying mutations from the SCZ patient showed more pronounced differences in the phenotypes when using the 3D spheroid models (spheroid size, cells scattered from the spheroid, protrusions appearing during differentiation). Our results emphasize the importance of cell adhesion in schizophrenia, while also highlighting the impact of individual background genetic variations on the in vitro cellular phenotypes [43].

Limitations of this study

The phenotypic heterogeneity of hiPSC lines and their derived cell types is well documented [44], and certain disease phenotypes may manifest with varying severity depending on the genetic background. Consequently, while it is beneficial to compare lines from multiple patients and healthy donors, examining isogenic lines is also advantageous due to their reduced genetic background variation. Although the present data show the importance of 3D cultures in studying neural development and the effects of drugs in this system, more comprehensive investigations with larger sample sizes are warranted in this regard. Also, the technology for the investigation of 3D-NPC behaviour should be improved. At the beginning of NPC differentiation (4 days in differentiation medium), the morphology of the processes and the number of scattered cells of the spheroids showed significant differences, but we could not statistically analyze these characteristics in detail. Because of the strong staining of the spheroids, the processes and the cell bodies on these processes cannot be separated from the background by the methods employed here. Additional software, developed by using artificial intelligence, may provide a solution in this regard, providing additional measuring conditions and parameters that are more suitable for further pharmacological studies (e.g., changes in the number of neurite branches). While differences in cell adhesion may play a role in disease development, it is important to recognize that the direct clinical translatability of these findings is limited and requires further investigation.

## 4. Materials and Methods

### 4.1. Generation of NPCs and Spheroids: Antipsychotic Treatments

#### 4.1.1. Cell Line Selection and Culture

For our experiments, we used an iPSC line derived from a schizophrenia patient carrying a de novo heterozygous mutation in the *ZMYND11* gene (Pat-Mut). As an isogenic control, we used the same cell line with the mutation corrected using CRISPR-Cas9 (Pat-Wt). Additionally, we included a healthy control cell line (Ctrl-Wt), in which we introduced the same mutation found in the patient with schizophrenia (Ctrl-Mut). Thus, we worked with a total of four cell lines (one clone for each case), described and characterized by our team in a previous article [10]. IPSC lines were differentiated into hippocampal neural progenitor cells (NPCs) and neurons. Hippocampal differentiation was performed using a published protocol [11], which we successfully applied multiple times [10,13,45]. NPCs were cultured in DMEM F12 media containing N2, B27, basic Fibroblast Growth Factor2 (bFGF2), and laminin under standard conditions (37 °C, 5% CO_2_). The medium was replaced every second day with fresh medium.

#### 4.1.2. Antipsychotic Treatment of NPCs

The selected antipsychotics, haloperidol (Sigma-Aldrich, St. Louis, MO, USA, PHR1724), risperidone (Sigma-Aldrich, R3030-10MG), and aripiprazol (Sigma-Aldrich, SML0935-10MG), were applied at two concentrations—indicated as high and low (Table 1). We analyzed the mean and effective plasma concentrations observed in patients (low), alongside a concentration that was an order of magnitude higher (high), as referenced in previous studies [46,47,48]. Because most antipsychotics do not have well-defined dose–response curves, higher concentrations were used to identify potential toxic effects and to account for the possibility that concentrations in the brain exceeding plasma levels may occur. All three compounds were dissolved in DMSO; therefore, the control was prepared using the solution with the highest DMSO content, and each treatment was adjusted to contain an equal amount of DMSO.

#### 4.1.3. Spheroid Formation

For 3D spheroid formation, NPCs were seeded (5 × 10^4^ cells per well) onto well ULA plates. Spheroids were cultured for three days to aggregate. On the third day, spheroid diameters were measured (*n* = 42) and the data were analyzed using the Kruskal–Wallis test followed by Dunn’s multiple comparisons test.

The aggregated spheroids were then transferred onto tissue culture plates coated with poly-L-ornithine and laminin. For the 3D migration and cell number experiments, we used the same medium as for NPC culturing, supplemented with antipsychotics and DMSO, and we cultured them for 3 days. For the 3D neurite outgrowth analysis, the spheroids were placed in a neural differentiation medium, also supplemented with the treatments, for four days. The differentiation media is N2, B27 containing DMEM F12 plus BDNF, WNT family member 3A (WNT3A), cAMP, and ascorbic acid.

#### 4.1.4. Two-Dimensional Proliferation and Scratch Assay

For the 2D experiments, the NPC medium was also supplemented with the drugs and DMSO. For the proliferation assay, 2 × 10^4^ cells per well were seeded onto a 96-well cell culture plate. Treatment began the day after seeding, and nuclei stained with DCV were imaged at 24, 48, and 72 h after treatment initiation (nine non-overlapping images per well). Due to staining, the medium was replaced daily.

For the 2D scratch assay, wells were allowed to reach confluence before performing a scratch wound. The medium was then replaced with the antipsychotic-supplemented medium, and images were taken 24 h after the scratch and treatment initiation.

### 4.2. RNAseq Data Analysis

In this study, we utilized previously generated RNA-seq data, which include samples from both the NPC and neuronal states across all four cell lines. The detailed description of RNA sequencing and data analysis can be found in our previous study [10].

### 4.3. DCV and Calcein-AM Staining

We used the Vybrant™ DyeCycle™ Violet (DCV) (Thermo Scientific, Waltham, MA, USA, V35003) chromogenic nuclear stain in the final concentration of 25 nM to visualize the cell nuclei (it refers to the cell number) in the case of 2D proliferation and 3D cell number. In the case of the 3D migration and 3D neurite experiments, we additionally stained the cytoplasm and the processes with Calcein-AM (Thermofisher, C3100MP) with a final concentration of 100 nM. The cells were incubated with both stains for 1 h, after which the medium was changed.

### 4.4. HCS Measurements and Analysis

For the determination of the cell number and neurite outgrowth, we used ImageXpress Micro XLS, a high-content screening device (Molecular Devices, San Jose, CA, USA) with the software provided by the company (Metaexpress software 64-bit version number is 5.3.0.5, Molecular Devices). Calcein staining was used to visualize the elongations, while DCV was used to visualize the nuclei. For each technical parallel, images were acquired from nine fields of view. For nuclei staining, the DAPI filter cube (ex. 377/50 nm, em. 447/60 nm); for calcein detection, the FITC filter cube (ex. 482/35 nm, em. 536/40 nm) was used with a 10× Nikon objective (Plan Fluor, NA = 0.3).

#### 4.4.1. Two-Dimensional Cell Number Determination

For the 2D nuclei count analysis, the Multi-Wavelength Cell Scoring Application Module of Metaexpress was used. The following parameters were set for the analysis of the pictures acquired with the DAPI filter: the minimum width of the nuclei was 9 µm, the maximum width was 15 µm, and the intensity above the local background was 250–550 arbitrary units (AU) based on the intensity of the DCV staining.

Since there is significant variation between the values from the nine non-overlapping images taken from the same well, we summed the values for each well, resulting in a single value per well. This approach provides a more representative data point for each well. For statistical analyses, the Kruskal–Wallis test and post hoc Dunn’s multiple comparisons test were performed to compare the various treatment effects in each treatment condition.

#### 4.4.2. Analysis of the Scratch Assay

For the determination of the 2D migration of cells, we used ImageJ software package (downloaded in 2024) and pictures of scratches, which were taken with a high-content screening microscope. All images from scratch were stacked following Image > Stacks > Images to stack. Following Image > Stacks > Make Montage and setting the number of images in the stack as “Columns” and “1” for “Rows”, with a scale factor of one, a full image of the scratch was created. The pixel width of the scratch was measured at five points using the Straight-Line tool. The measurement points were chosen to represent the average width the best. After each measurement, overlays of each line and montaged pictures were saved. For the analysis, the average length of the five measurement lines was calculated. The difference in scratch sizes after 24 h was analyzed by one-way ANOVA, Šídák’s multiple comparisons test (*n* = 7–17).

#### 4.4.3. Three-Dimensional NPC Number Determination

For 3D NPC number determination, a custom macro was developed on ImageJ. The macro ensures standardization and quantification of image processing. The images are converted to 8-bit grayscale format, and the contrast is enhanced with 0.35% saturation by the macro. The “Threshold” algorithm of ImageJ is used with default settings. The images are converted to binary masks after ensuring the background is black and cells are represented as white particles. Using the “Analyze Particles” algorithm of ImageJ, the measurements regarding count, total area, average size, and %area values are obtained. For more details, see Appendix A. These values were used for the characterization of the outgrowth of spheroids. Cell numbers on the spheroid outgrowth on the third day were analyzed by *n* = 4–6, using one-way ANOVA and Šídák’s multiple comparisons test. (*n* = 4–6 in Figure 4b). Cell numbers on the spheroid outgrowth on the third day after antipsychotic treatment were analyzed by the Krsukal–Wallis test and Dunn’s multiple comparisons test (*n* = 4–6 in Figure 5).

#### 4.4.4. Three-Dimensional NPC Outgrowth Analysis

For the determination of the migration of cells in spheroids, we used ImageJ and pictures of each spheroid. All spheroid images from a single well were stacked following Image > Stacks > Images to stack. To measure the inner area of spheroids, a Gaussian blur with a sigma (radius) value was set to 5.00. Following the blurring, images were converted to masks by following Process > Binary > Convert to Mask when “Method” and “Background” settings were set to Default, and “Calculate threshold for each image” and “Black background (of binary masks)” options were marked. Before measurement, Analyze > Set Measurements was configured to include “Area”, “Feret’s diameter”, and “Display label”. The area of the spheroid was measured in pixels using the Wand (tracing) tool, and the white region in the image was selected. All images’ measurements, masked versions, and ROI data were saved. To measure the outgrowth of spheroids, the outer region where cells are dense was selected using the Freehand Selections tool. Measurement results in pixels were obtained following the same measurement settings as the inner shell. The measurements and overlays of selections were saved for each picture. Migration analysis was performed by determining the radii of the inner and outer parts, assuming that they were circular. Spheroid outgrowth on the third day was analyzed by one-way ANOVA and Šídák’s multiple comparisons test (*n* = 4–6 in Figure 4c). Statistical analysis of spheroid outgrowth on the third day after antipsychotic treatment used one-way ANOVA and Dunnett’s multiple comparisons test (*n* = 4–6 in Figure 6).

#### 4.4.5. Determination of Neurite Outgrowth in 3D Differentiated NPCs

For the determination of the neurite outgrowth in differentiated 3D NPCs, ImageJ and the images of the spheroids were used. We measured the longest, farthest-reaching neurite’s distance by connecting the neurite’s endpoint to the spheroid’s centre with a straight line, using the built-in *Line tool* of ImageJ. Next, by lengthening this line to the opposite side of the spheroid, we measured the distance of the farthest neurite intersecting this line. Finally, we drew two perpendicular lines and measured the length of the furthest neurites on each side. The result was a cross of four lines with 90-degree angles between them. Each image’s four neurite lengths were saved in pixels. Neurite outgrowth in 3D differentiated NPCs was analyzed by the Kruskal–Wallis test and Dunn’s multiple comparisons test (*n* = 4–6). Statistical analysis of the outgrowth of differentiating spheroids treated with different antipsychotics on the 4th day were analyzed by the Kruskal–Wallis test and post hoc Dunn’s multiple comparisons test (*n* = 3–6).

## Figures and Tables

**Figure 1 ijms-26-04444-f001:**
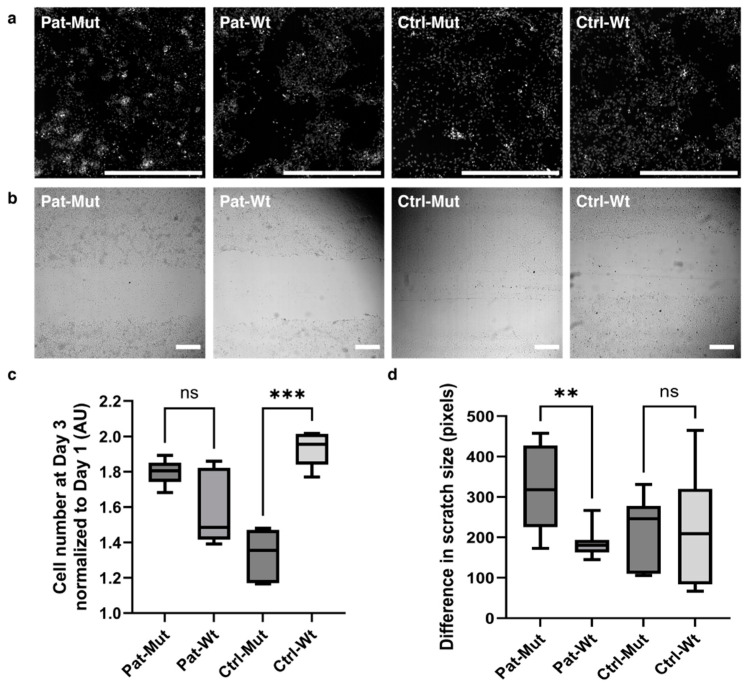
Proliferation capacity and migration of NPCS in 2D: (**a**) Representative pictures of NPC cultures, the cell nuclei were stained by DCV. The scale bars represent 800 µm. (**b**) Representative bright field pictures of scratch assays in different NPC cultures (see also Appendix A). The scale bars represent 500 µm. (**c**) Statistical analysis of the change in cell number on day 3. The normalized cell numbers are shown as average ± SD. *n* = 6–9 analyzed by the Kruskal–Wallis test and post hoc Dunn’s multiple comparisons test. (**d**) Statistical analysis of the difference in scratch sizes after 24 h. The reduction in scratch size is shown in pixels as average ± SD. *n* = 7–17 analyzed by one-way ANOVA, Šídák’s multiple comparisons test, ** *p* ≤ 0.01, *** *p* ≤ 0.001, ns = non-significant.

**Figure 2 ijms-26-04444-f002:**
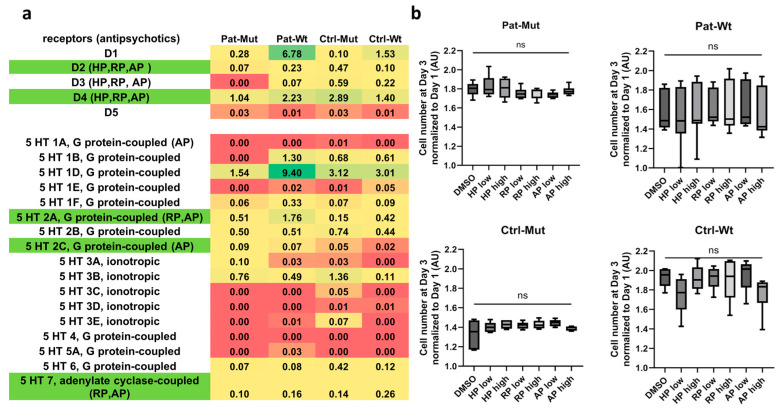
Characterization of receptor expression based on mRNAseq data and effect of antipsychotics on cell proliferation (**a**) mRNA expression of dopamine and serotonin receptors in NPCs. The antipsychotic that acts on the given receptor has been indicated in parentheses. The receptors on which antipsychotics act and are expressed in the in vitro system are highlighted in green. The expressions are indicated in Reads Per Kilobase Million (RPKM). (**b**) Statistical analysis of the change in cell number of the treated cells on day 3. Cell counts are shown as average ± SD. *n* = 6–9 analyzed by Kruskal–Wallis test and post hoc Dunn’s multiple comparisons test, ns = non-significant.

**Figure 3 ijms-26-04444-f003:**
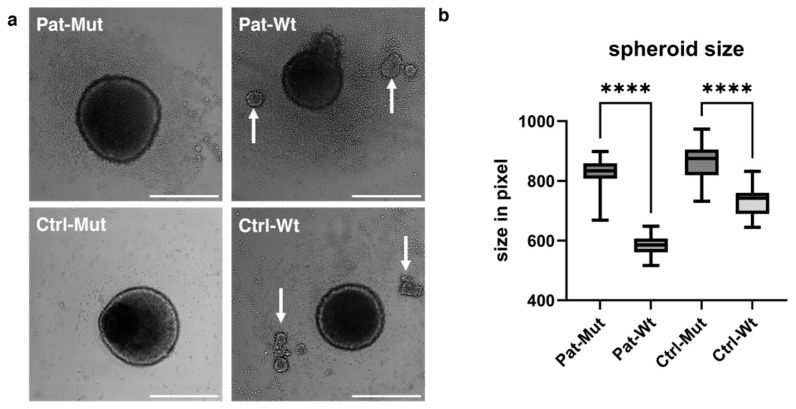
Morphological characterization of spheroids. (**a**) Representative pictures of spheroids. The white arrows indicate the scattered smaller spheroids in the Wt samples. The scale bars represent 500 µm. (**b**) Statistical analysis of spheroid size on the 3rd day. The spheroid size is shown in pixels as average ± SD. *n* = 42 analyzed by Kruskal–Wallis test and Dunn’s multiple comparisons test, **** *p* ≤ 0.0001.

**Figure 4 ijms-26-04444-f004:**
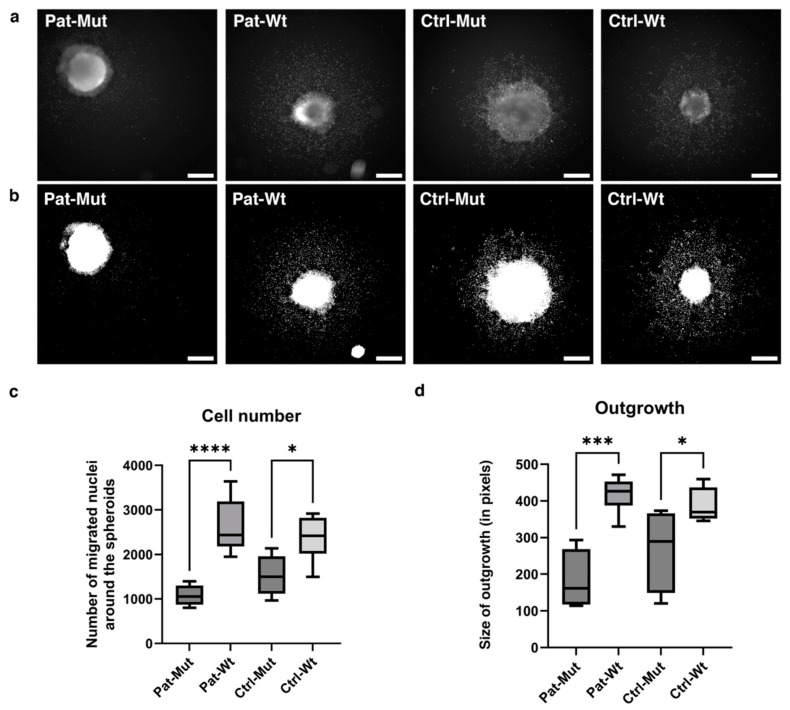
Characterization of the outgrowth of the spheroids. (**a**,**b**) Representative pictures of spheroids after 3 days of attachment. The upper images show the nuclei stained with DCV (**a**), and the lower ones show the mask (**b**) applied by the algorithm (see Section 4 for details). The scale bars represent 500 µm. (**c**,**d**) Statistical analysis of spheroid outgrowth on the 3rd day. The number of migrated cells and the size of outgrowth (in pixels) are shown as average ± SD. *n* = 4–6 analyzed by one-way ANOVA and Šídák’s multiple comparisons test, * *p* ≤ 0.05, *** *p* ≤ 0.001, **** *p* ≤ 0.0001.

**Figure 5 ijms-26-04444-f005:**
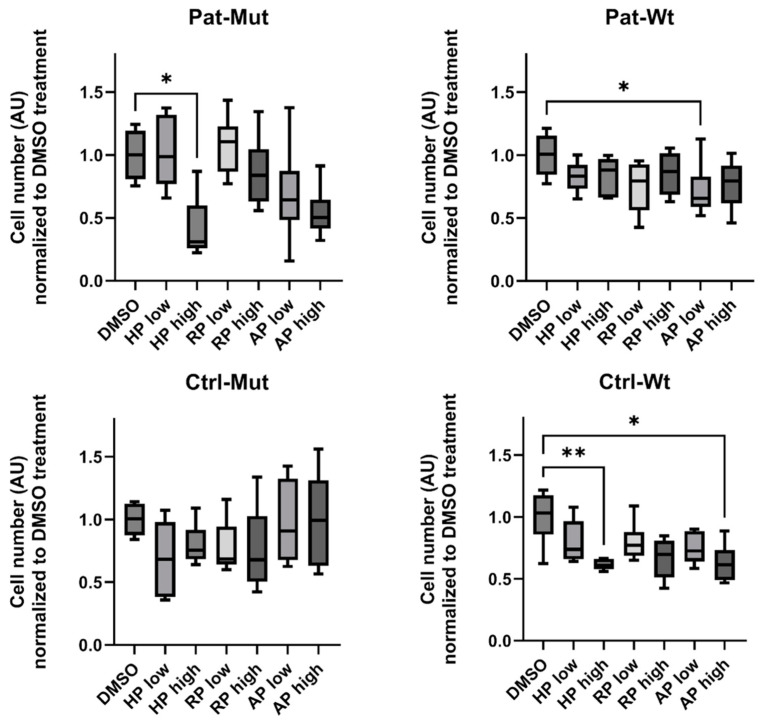
Characterization of cell numbers on the outgrowth of the treated spheroids. Statistical analysis of cell numbers on the spheroid outgrowth on the 3rd day of treatment. The number of migrated cells (normalized to DMSO-treated cells) is shown as average ± SD. *n* = 4–6 analyzed by Kruskal–Wallis test and Dunn’s multiple comparisons test, * *p* ≤ 0.05, ** *p* ≤ 0.01.

**Figure 6 ijms-26-04444-f006:**
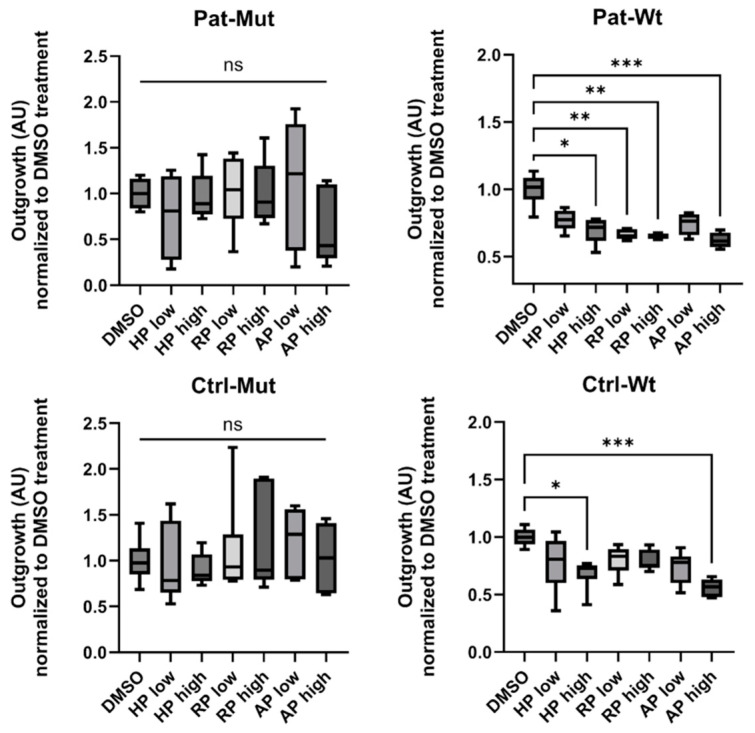
Characterization of the outgrowth of the treated spheroids. Statistical analysis of spheroid outgrowth on the 3rd day of treatment. The size of the outgrowth (normalized to DMSO-treated cells) is shown as average ± SD. *n* = 4–6 analyzed by Ordinary one-way ANOVA and Dunnett’s multiple comparisons test, * *p* ≤ 0.05, ** *p* ≤ 0.01, *** *p* ≤ 0.001, ns = non-significant.

**Figure 7 ijms-26-04444-f007:**
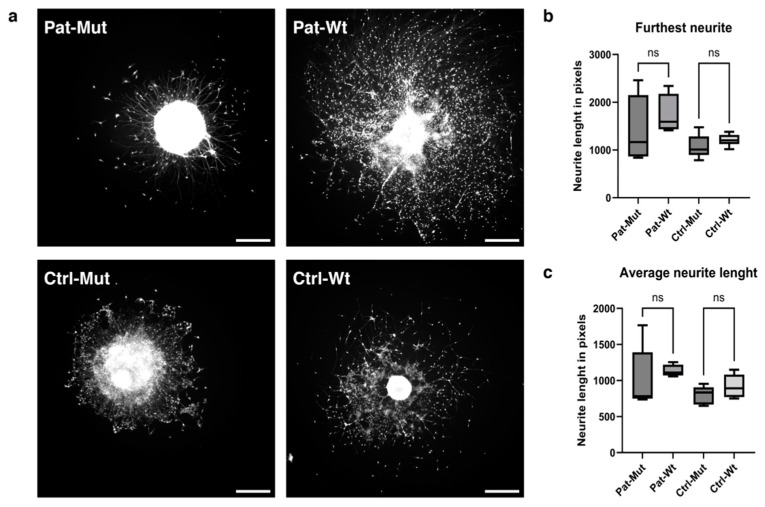
Characterization of the outgrowth of differentiating spheroids. (**a**) Representative pictures of differentiating spheroids. The cells were stained with Calcein-AM. The scale bars represent 500 µm. (**b**,**c**) Statistical analysis of spheroid outgrowth on the 4th day after attachment (see Section 4 for details). The characteristics of neurite length are shown in pixels as average ± SD. *n* = 4–6 analyzed by Kruskal–Wallis test, Dunn’s multiple comparisons test, ns = non-significant

**Figure 8 ijms-26-04444-f008:**
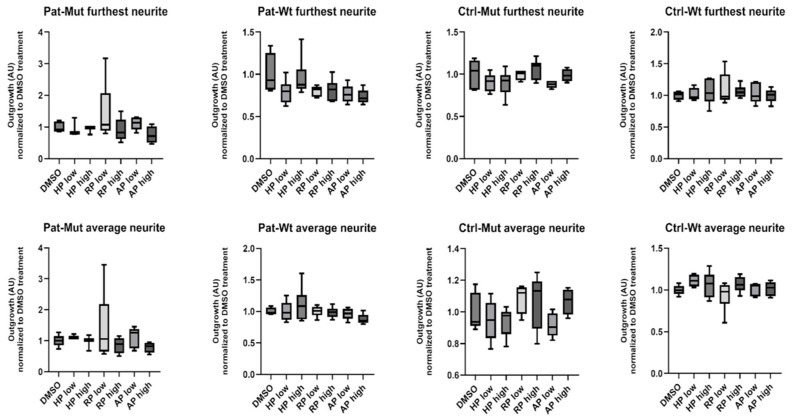
Characterization of the outgrowth of differentiating spheroids treated with different antipsychotics. Statistical analysis of spheroid outgrowth on the 4th day of treatment (see Section 4 for details). The characteristics of neurite length are shown in pixels as average ± SD. *n* = 3–6 analyzed by Kruskal–Wallis test and Dunn’s multiple comparisons test.

**Figure 9 ijms-26-04444-f009:**
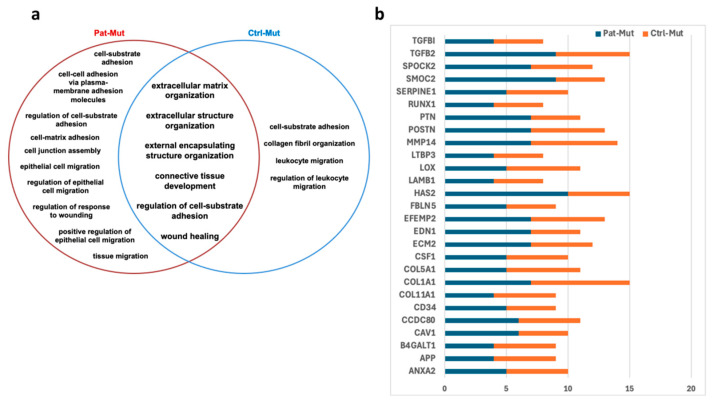
Characterization of cell adhesion based on mRNAseq data: (**a**) Distribution of GO terms related to cell adhesion in the mutation carrying NPCs depicted in a Venn diagram. (**b**) The most frequently occurring UE genes in GO terms related to cell adhesion in NPCs carrying the *ZMYND11* mutation. The number of occurrences in Pat-Mut (blue) and Ctrl-Mut (orange) is shown.

**Table 1 ijms-26-04444-t001:** Summary of antipsychotic concentrations used.

Group	Treatment	Concentration	
HPlow	Haloperidol	10 ng/mL	0.003 µM
HPhigh		100 ng/mL	0.03 µM
RPlow	Risperidone	100 ng/mL	0.24 µM
RPhigh		1000 ng/mL	2.4 µM
APhigh	Aripiprazole	100 ng/mL	0.22 µM
APlow		1000 ng/mL	2.2 µM
DMSO	DMSO	0.2 µL/mL	

## Data Availability

The data underlying this article are available on reasonable request.

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
