# Peer review of "Functional Analysis of Antipsychotics in Human iPSC-Based Neural Progenitor 2D and 3D Schizophrenia Models"

_ijms, 2025, doi:10.3390/ijms26094444_

Round 1
Reviewer 1 Report
Comments and Suggestions for Authors
The study presents a comprehensive investigation into the effects of antipsychotics on 2D and 3D neural progenitor cell (NPC) models derived from iPSCs with ZMYND11 mutations associated with schizophrenia. The integration of CRISPR-edited isogenic controls and the exploration of both 2D and 3D culture systems are notable strengths, offering insights into cell adhesion-related phenotypes and differential drug responses. However, several methodological and interpretative limitations need to be addressed to strengthen the validity and translational relevance of the findings.
Major Comments:
- While the 3D spheroid model revealed significant differences in spheroid size and outgrowth between mutant and wild-type NPCs, a lack of quantitative analysis for neurite branching and scattered cell numbers during differentiation.
- The rationale for selecting drug concentrations (Table 1) based on "known plasma concentrations of patients" is unclear.
- While the study uses CRISPR-corrected and mutant lines, no data is provided to confirm the functional rescue or introduction of the ZMYND11 mutation. Without this validation, phenotypic differences between Pat-Mut/Pat-Wt and Ctrl-Mut/Ctrl-Wt cannot be definitively attributed to the ZMYND11 mutation.
- The mRNA-seq data highlight downregulation of adhesion-related genes (e.g., TGFB2, COL1A1), but functional assays (e.g., ECM composition analysis, adhesion molecule expression) are absent.
- The discussion should elaborate on the implications of genetic background variability.
Minor Comments:
- Figures 1a, 3a, and 7a lack sufficient resolution to discern cellular details. Authors should provide higher-magnification images or alternative staining.
- Move critical supplementary data (e.g., GO term Venn diagrams, mRNA-seq receptor expression profiles) to the main figures to directly support claims about adhesion mechanisms and antipsychotic targets.
Reviewer 2 Report
Comments and Suggestions for Authors
The work covers a relatively uninvestigated area of schizophrenia research, i.e., the impact of antipsychotics on neural progenitor cells (NPCs) in 2D and 3D systems. This is a new and timely perspective. The interpretation of findings, especially with regard to differential drug responses, however, requires more mechanistic explanation. Although the manuscript is detailed, it reads in many places like a data report rather than an integrated scientific narrative. Try to guide the reader more directly through the logic, especially when transitioning between 2D and 3D models or comparing between different cell lines. The conclusion states that cell adhesion differences "may contribute to disease development." This is too bold based on the current in vitro data alone. Attempt to soften this to reflect the limitation of direct clinical translatibility.
1. The utilization of both corrected and CRISPR-engineered lines is a plus. Nevertheless, more specific definition of the numbers of clones utilized for each condition would make it easier to gauge biological variation.
2. Authors discuss software limitations for quantifying neurite outgrowth and complexity. This is an important caveat, but it reduces confidence in conclusions made from Figures 7-8. Attempt to confirm the morphology findings with a more sensitive assay or AI-based image analysis software.
3. Figure 2A shows dopamine and serotonin receptor expression, but not in comparison with expression in mature neurons or to in vivo controls. Are the levels physiologically relevant? A few sentences of discussion would be helpful.
4. The fact that mutant spheroids are larger but show reduced outgrowth is interesting. Could the increased size reflect changes in adhesion, proliferation, or just different compaction dynamics? A more specific hypothesis would be helpful.
5. Although stated to be derived from known plasma concentrations, a table comparing these with in vivo CSF or brain levels (where available) would provide additional validation.
6. The authors mention that differences in background could explain phenotypic variation beyond the ZMYND11 mutation. Although the observation is important, the discussion would benefit from clearer acknowledgment of the limitations involved in interpreting causality in comparisons between non-isogenic and isogenic entities.
7. The study focuses on progenitor stages, yet antipsychotics are typically addressed in the framework of adult neuronal dysfunction. While the rationale is reasonable, additional explanation of how early NPC reactions might be connected to later brain alterations in schizophrenia would be helpful.
8. There is a good emphasis on cell adhesion genes (e.g., TGFB2, COL1A1, MMP14). However, the functional relationship between these expression changes and subsequent phenotypes (migration, spheroid detachment) can be strengthened by functional validation.
9. Some of the spheroid photographs (e.g., Fig. 3a and 4a) are low contrast and poorly labeled. Adding labels directly on the figure (e.g., "Pat-Mut", "Ctrl-Wt") would facilitate reading.
10. The majority of estimates have pertinent statistics but small sample sizes (n = 4-6 in most cases). This needs to be noted as a limitation in the primary discussion.
11. Several of the figure legends (e.g., Fig. 5, 6, 8) are extremely short and could be longer to describe treatment conditions, timepoints, and specific statistical tests used.
12. There are some minor typographical errors in the manuscript, such as missing spaces and lack of consistency in formatting (e.g., using "SCZ-s" instead of "SCZs" in some places). A careful proofreading would help.
Round 2
Reviewer 1 Report
Comments and Suggestions for Authors
I have no more comments.
Reviewer 2 Report
Comments and Suggestions for Authors
The revised manuscript reflects improvements made in response to the reviewers’ comments.